# Host-parasite coevolution promotes innovation through deformations in fitness landscapes

**Animesh Gupta[1], Luis Zaman[2], Hannah M Strobel[3], Jenna Gallie[4], Alita R Burmeister[5], Benjamin Kerr[6], Einat S Tamar[7], Roy Kishony[7]\*, Justin R Meyer[3]\***

[1]Department of Physics, University of California San Diego, La Jolla, United States; [2]Department of Ecology and Evolutionary Biology, University of Michigan, Ann Arbor, United States; [3]Department of Ecology, Behavior and Evolution, University of California San Diego, La Jolla, United States; [4]Department of Evolutionary Theory, Max Planck Institute for Evolutionary Biology, Plön, Germany; [5]Department of Ecology and Evolutionary Biology, Yale University, New Haven, United States; [6]Department of Biology, University of Washington, Seattle, United States; [7]Department of Biology, Technion – Israel Institute of Technology, Haifa, Israel

**\*For correspondence:**
rkishony@tx.technion.ac.il (RK);
jrmeyer@ucsd.edu (JRM)

**Competing interest:** The authors declare that no competing interests exist.

**Abstract** During the struggle for survival, populations occasionally evolve new functions that give them access to untapped ecological opportunities. Theory suggests that coevolution between species can promote the evolution of such innovations by deforming fitness landscapes in ways that open new adaptive pathways. We directly tested this idea by using high-throughput gene editing-phenotyping technology (MAGE-Seq) to measure the fitness landscape of a virus, bacteriophage $\lambda$, as it coevolved with its host, the bacterium *Escherichia coli*. An analysis of the empirical fitness landscape revealed mutation-by-mutation-by-host-genotype interactions that demonstrate coevolution modified the contours of $\lambda$'s landscape. Computer simulations of $\lambda$'s evolution on a static versus shifting fitness landscape showed that the changes in contours increased $\lambda$'s chances of evolving the ability to use a new host receptor. By coupling sequencing and pairwise competition experiments, we demonstrated that the first mutation $\lambda$ evolved en route to the innovation would only evolve in the presence of the ancestral host, whereas later steps in $\lambda$'s evolution required the shift to a resistant host. When time-shift replays of the coevolution experiment were run where host evolution was artificially accelerated, $\lambda$ did not innovate to use the new receptor. This study provides direct evidence for the role of coevolution in driving evolutionary novelty and provides a quantitative framework for predicting evolution in coevolving ecological communities.

## Editor's evaluation

This study uses the parlance and framing of the fitness landscape to articulate a co-evolution story between host and parasite. It utilizes a tractable system, bacteriophage $\lambda$ and *E. coli*, to ask questions that unite different pillars of evolutionary theory – evolutionary genetics (via the fitness landscape analogy), co-evolution, and host-parasite interactions. The findings will be relevant to a number of audiences, and will likely spawn downstream studies that further interrogate the molecular specifics that underlie host-parasite co-evolution.

## Introduction

A starting point for understanding how populations evolve is to assume that they exist in an unchanging world where they can adapt toward optimality (*Orr, 2005*; *Pigliucci and Müller, 2010*). However, even in static environments, populations never reach optimality because their circumstances continuously change as neighboring species coevolve with them (*Valen, 1973*). This more dynamic view of the evolutionary process opens the potential for unbounded evolution and creates new opportunities for evolutionary innovation (*Doebeli, 2011*; *Thompson, 2005*; *Nahum et al., 2017*; *Thompson and Cunningham, 2002*; *Zaman et al., 2014*). Darwin recognized this potential in the final pages of On the Origin of Species, where he wrote that, 'It is interesting to contemplate an entangled bank' of organisms evolving with one another to produce such a variety of forms and functions (*Darwin, 1859*). But he also realized the empirical challenges created by the richness of species interactions within ecological communities in his further description of 'these elaborately constructed forms, … dependent on each other in so complex a manner…' (*Darwin, 1859*). The complexity arises because an organism's fitness is a function of its interactions with other species, and the strength and form of these interactions can continuously change as they coevolve. Furthermore, the coevolving traits of organisms are encoded within genomes by mutations that might interact with one another, a pervasive phenomenon called epistasis (*Weinreich et al., 2006*). This means that interactions at all levels must be considered; from mutation-by-mutation within a species (classical epistasis), to mutation-by-mutation between species (interspecific epistasis), and higher order phenomena such as the combination of classic and interspecific epistasis, where the within genome mutation-by-mutation interactions depend on the genotype of an interacting species.

Many advances have been made over recent decades that enable us to tackle this combinatorial problem. Efficient genetic engineering methods permit the construction of genetic libraries with combinatorial sets of mutations that can be used to measure epistasis (*Kosuri and Church, 2014*; *Fowler and Fields, 2014*). Also available are convenient approaches to measure Darwinian fitness of the mutant libraries (*Weinreich et al., 2006*; *Palmer et al., 2015*; *Khan et al., 2011*; *Chou et al., 2011*). Coupling these two technologies allows the creation of extensive genotype-to-fitness maps, or fitness landscapes (*Wright, 1932*), that provide information important for predicting adaption (*de Visser and Krug, 2014*; *de Visser et al., 2018*; *Lee et al., 2018*). However, these maps alone are often not sufficient to predict evolution because their topographies can depend on abiotic environmental conditions (*Ogbunugafor et al., 2016*; *Lindsey et al., 2013*; *Steinberg and Ostermeier, 2016*; *Flynn et al., 2013*) and biotic interactions (*Cervera et al., 2016*; *Fragata et al., 2019*). Here, we take two significant steps forward in fitness landscape research. First, we build on the observation that landscape structures depend on species interactions by studying the interdependence of two species' landscapes and how they shift during coevolution. Second, we test whether these shifts facilitate the evolution of a key innovation, where the species evolves a new function that unlocks new ecological opportunities.

As a model system of coevolution, we studied the host-parasite interaction between bacteriophage $\lambda$ and its host, *Escherichia coli*, because of the extensive background research completed on their coevolution and the availability of well-developed molecular tools (*Meyer et al., 2012*; *Maddamsetti et al., 2018*). When $\lambda$ and *E. coli* are cocultured in the laboratory, one quarter of the $\lambda$ populations evolve to use a new receptor (*Meyer et al., 2012*). $\lambda$'s native receptor is *E. coli*'s outer-membrane protein LamB, but through mutations in its host-recognition gene *J*, $\lambda$ evolves to use a second receptor protein, OmpF. While only four mutations are necessary for OmpF+ function (*Maddamsetti et al., 2018*), more *J* mutations typically evolve along the way (*Meyer et al., 2012*). $\lambda$ gains this new function after *E. coli* evolves resistance through *malT* mutations (*Meyer et al., 2012*) that cause reduced LamB expression (*Boos and Böhm, 2000*). Thus, it was hypothesized that the evolution of resistance in *E. coli* deformed $\lambda$'s fitness landscape in ways that promoted $\lambda$'s innovation (*Thompson, 2012*). In line with this, it was previously shown that four out of six $\lambda$ genotypes randomly chosen on the path to evolve OmpF+ had higher relative fitness when cultured with resistant *malT⁻* cells rather than ancestral cells (*Burmeister et al., 2016*), suggesting that the host's coevolution would promote key steps in $\lambda$'s evolution. However, one out of six genotypes had higher fitness in the presence of the ancestral host, and the last $\lambda$'s fitness was neutral to the host's genotype. Given the conflicting pattern and small sample size, it could not be concluded whether coevolution was responsible for the innovation. Here, we build on this study with high-throughput technologies capable of

measuring the fitness of hundreds of $\lambda$ genotypes. The technology also produces combinatorial-mutation libraries of $\lambda$ genotypes that can be used to quantify epistasis. This allows us to establish the contours of $\lambda$'s adaptive landscape rather than simply studying isolated genotypes within the space. Given the efficiency of this method, we can now measure $\lambda$'s landscape repeatedly in different host contexts in order to test whether host-induced deformations that naturally arise during coevolution promote OmpF$^+$ evolution. Although it has been shown that antagonistic coevolution can hasten molecular evolution of phages (*Paterson et al., 2010*) and lead them to broader host ranges (*Hall et al., 2011*), coevolution's role in unlocking unexplored regions of the fitness landscape has not been directly tested.

## Results

### λ's fitness landscape at different stages of coevolution

To construct $\lambda$'s fitness landscape, we focused on 10 *J* mutations that were a subset of mutations $\lambda$ repeatedly evolved on its path to use OmpF (*Meyer et al., 2012*; *Supplementary file 1a*). Together, they form a 10-dimensional genotype space with a total of 1024 (2$^{10}$) unique variants of different combinations of the mutations including, the wild type (WT) allele configuration. Using Multiplexed Automated Genome Engineering (MAGE) (*Wang et al., 2009*), a technique that uses repeated cycles of homologous recombination in the $\lambda$-red system to produce combinatorial genomic diversity, we were successful at engineering a library of 671 genotypes out of the possible 1024 (see Materials and methods). To measure the fitness of each genotype in this library, we competed the full library en masse and monitored their frequency changes using next-generation sequencing (*Figure 1—figure supplement 1*; *Kelsic et al., 2016*; *Russ et al., 2020*). The fitness of each genotype was then calculated by comparing its change in frequency relative to the non-engineered ancestor. Fitness was measured in four replicate competitions for both the ancestral host and *malT*$^-$ host (see Materials and methods). To reduce the effect of sequencing errors and to overcome other methodological pitfalls, we modified the MAGE protocol by introducing neutral *watermark* mutations in the library construction and developed a high-throughput competition assay that yielded reproducible results (see Materials and methods, *Figure 1—figure supplement 2*, Appendix 1). Overall, we were able to measure the fitness of 580 $\lambda$ genotypes cocultured with ancestral *E. coli* and 131 genotypes with *malT*$^-$ (*Figure 1—figure supplement 3*). The reduced number of genotypes compared to the initial library was due to a combination of factors. The randomness of the MAGE editing process caused some genotypes to be rarely constructed and to have low frequencies in the initial library. If these genotypes did not possess disproportionately high fitness, then their frequency would fall below the limit of detection during the competition, removing them from the analysis. This effect was more pronounced in the *malT*$^-$ landscape, where fitness differences were even more extreme.

Visual inspection of the two fitness landscapes reveals host-dependent structures; the landscape with the ancestral host has a standard diminishing-returns pattern (*Khan et al., 2011*; *Chou et al., 2011*; *Kryazhimskiy et al., 2014*; *MacLean et al., 2010*; *Guerrero et al., 2019*), while the landscape with the *malT*$^-$ host has an atypical sigmoidal shape that plateaus at a higher fitness than the first (*Figure 1a and b*). The non-linear relationship between mutation number and fitness suggests the presence of epistasis (mutation-by-mutation interactions), the differences in the magnitude of fitness effects between landscapes suggests mutation-by-host interactions, and different shapes suggest host-dependent epistasis (mutation-by-mutation-by-host interactions). To determine how much variation in fitness is explained by these interactions, we performed multiple linear regression analyses (see Materials and methods). We found pervasive epistasis in both landscapes (*Figure 1c*). For the ancestral landscape, 58.66% of the variation was explained by the direct effects of mutations and 24.69% by pairwise interactions ($R^2_{adj} = 0.8172$, $F_{55,439} = 39.97$, p<0.0001). Similarly, 48.35% of the variance in the *malT*$^-$ landscape was explained by the direct mutation effects and 27.61% by the interaction terms ($R^2_{adj} = 0.7072$, $F_{55,252} = 14.48$, p<0.0001). To test for mutation-by-mutation-by-host interactions, we regressed another linear model that includes host as a predictor variable. In this model, we found significant mutation-by-host interactions (*Figure 1d*), and sizeable effects of the host-dependent epistasis (21 mutation-by-mutation-by-host interaction terms were significant out of 45 and 12.62% of the total variance in the data were attributable to these terms, *Figure 1d*). This three-way interaction term

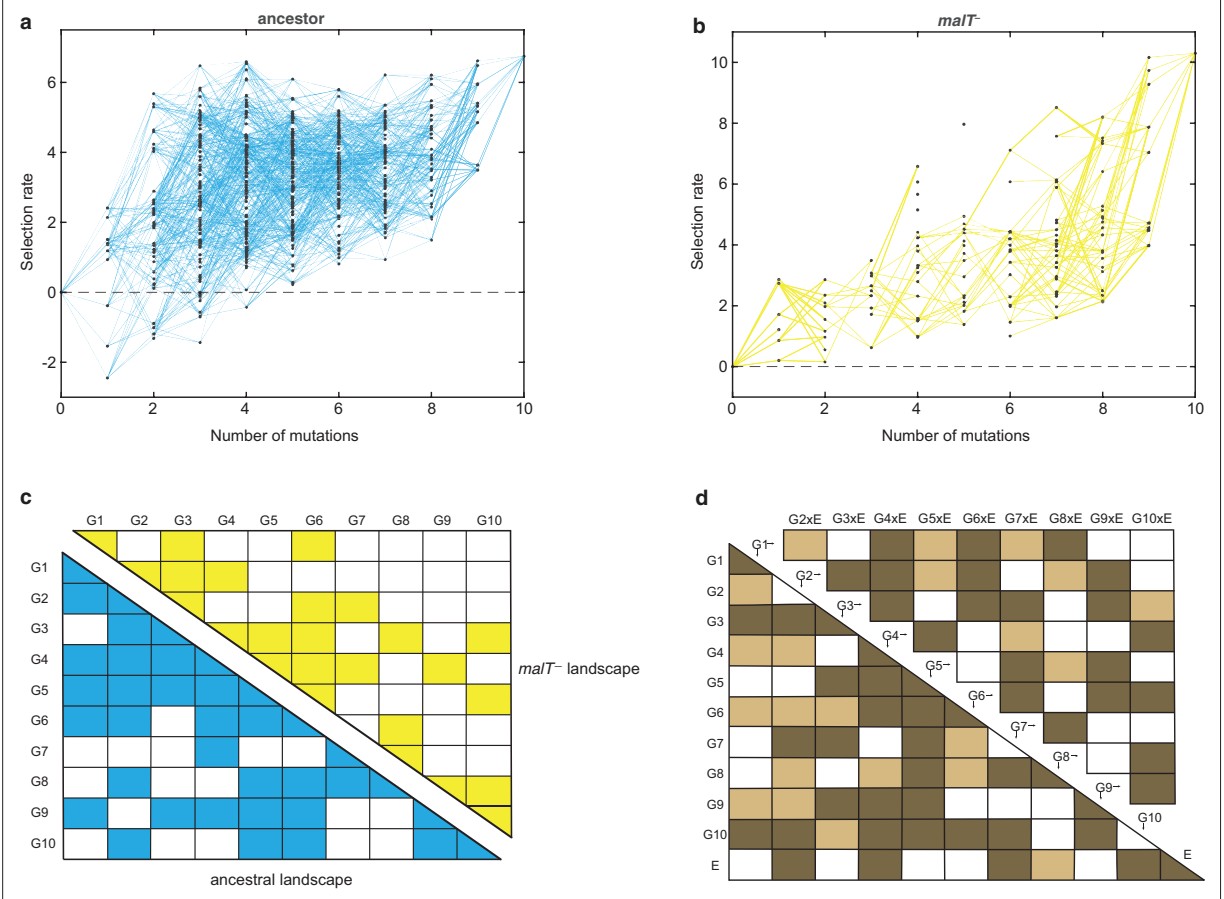

**Figure 1.** Empirical fitness landscapes of $\lambda$ when infecting the (**a**) ancestral host and (**b**) *malT⁻* host, and their statistical analyses in (**c**) and (**d**). Each node in (**a**) and (**b**) represents a unique genotype and two nodes are connected by edges if the corresponding genotypes are separated by one mutation. The node at zero mutations is ancestral $\lambda$. Selection rate (per 4 hr competition experiment) is the difference of Malthusian growth rates of a given genotype $i$ to ancestral $\lambda$ over 4 hr, calculated as $\ln\left(\lambda_{i,\,4}\lambda_{i,0}\right) - \ln\left(\lambda_{anc,\,4}\lambda_{anc,0}\right)$, where $\lambda_{i,\,t}$ denotes the density of the given genotype at time $t$. (**c**) Statistical analysis of direct and interactive effects of mutations in both the landscapes. Colored cells represent statistically significant terms determined by multiple regression analysis after correction for multiple hypothesis testing (see Materials and methods). The diagonal elements of the matrix represent single mutation effects and all the off-diagonal terms represent pairwise epistatic interactions. See ***Supplementary file 1i*** for identity of mutations corresponding to different $G_i$. (**d**) Statistical test of whether the two landscapes varied in topology. The additional variable, $E$, represents environment (host) to indicate mutation-by-host effects in the lower-left matrix and mutation-by-mutation-by-host (G×G×E) in the upper-right matrix. Light colored cells indicate terms present in the final AIC-optimized model out of the full-factorial model ($F_{76,726} = 37.45$, p<0.0001), and dark colored cells indicate statistically significant terms after controlling for rate of false positives (see Materials and methods).

The online version of this article includes the following figure supplement(s) for figure 1:

**Figure supplement 1.** Schematic illustration of MAGE-Seq used to construct $\lambda$'s fitness landscapes.

**Figure supplement 2.** Test to determine whether amplification distorts measurements of fitness.

**Figure supplement 3.** Distribution of $\lambda$ genotypes present with respect to the number of mutations they possess.

**Figure supplement 4.** Test of whether the error in our estimates of selection rates (per 4 hr) is influenced by the magnitude of the estimate.

measures the extent to which the landscape structure is transformed by host evolution and suggests that $\lambda$'s evolutionary trajectory could depend on its host's genotype.

## The role of shifting landscapes in promoting λ's innovation

To test whether changes in the structure of $\lambda$'s landscape opened trajectories to OmpF exploitation, we simulated $\lambda$'s evolution on the landscapes using a modified Wright-Fisher model (see Materials and methods, *Figure 2—figure supplement 1*). Before running the simulations, we imputed the missing $\lambda$ genotypes' fitness values to complete the landscapes. We did this by successively choosing missing genotypes at random and assigning them the average fitness of their nearest neighbors.

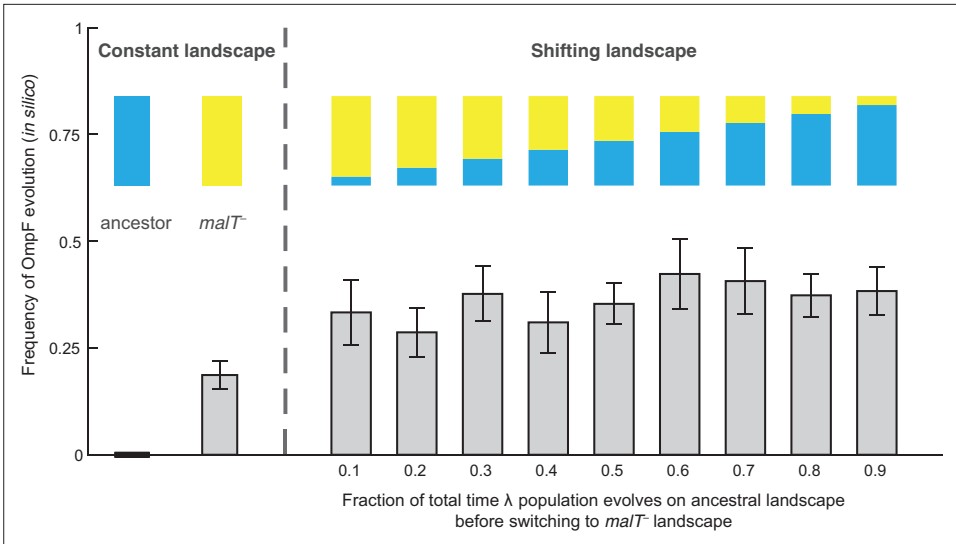

**Figure 2.** Simulation results of the frequency of OmpF-use evolution observed when fitness landscapes were shifted at different frequencies. Each bar represents an average of 300 simulation runs. Error bars indicate 95% confidence intervals. OmpF evolution is favored when $\lambda$ evolves on shifting landscapes. The only two shifting landscape treatments that are not significantly higher than simulations on the constant $malT^-$ landscape are the 0.2 and 0.4 treatments (*Supplementary file 1b*).

The online version of this article includes the following figure supplement(s) for figure 2:

**Figure supplement 1.** Schematic outline of the simulations used to evolve $\lambda$ on the fitness landscapes.

**Figure supplement 2.** Additional simulations to verify that shifting landscapes promote OmpF⁺ evolution.

The simulations were run based on conservative estimates of the number of generations and population sizes from previously published (*Meyer et al., 2012*) and this study's coevolution experiments (960 generations; ~6.3×10⁹ $\lambda$ particles, *Figure 4—figure supplement 1*; *Meyer et al., 2012*) and $\lambda$'s intrinsic mutation rate (7.7×10⁻⁸ base⁻¹ replication⁻¹) (*Drake, 1991*). We predicted that $\lambda$ would be more likely to evolve OmpF function (three specific mutations plus one additional *Maddamsetti et al., 2018*) in simulations that accounted for coevolution by shifting the population from one landscape to the next. We ran trials where $\lambda$ evolved on only one landscape at a time to establish a baseline for the frequency of OmpF⁺ evolution without coevolution. Next, we ran nine shifting landscape scenarios where we varied how many generations $\lambda$ evolved on the ancestral host landscape before switching to the *malT⁻* landscape. As anticipated, the switching protocol increased the frequency of OmpF⁺ evolution in all nine treatments above the single host simulations, but only seven out of nine treatments were found to be significantly higher (*Figure 2*; ANOVA: F-ratio=6.14, *d.f.*=99, p<0.0001, *Supplementary file 1b*). This result was robust to changes in population size and total number of generations, and when controlling for both, different number of genotypes measured in the two landscapes, and noise created by imputing missing data points (*Figure 2—figure supplement 2*, Appendix 2).

## Reconstructing coevolution in an experimental population

The simulation results suggest that the shifting landscape encourages $\lambda$'s evolution to gain the required mutations for OmpF function. In particular, the simulations show that the first steps along the path to the innovation are more likely if $\lambda$ first adapts to the ancestral bacterium, meanwhile the final steps are more likely to occur if the host coevolves resistance. To verify this result with laboratory experiments, we analyzed the path $\lambda$ took to OmpF⁺ in a single population cryopreserved from the previous coevolution study (population 'D7' in *Meyer et al., 2012*; Table S1). Population 'D7' was chosen because $\lambda$ evolved relatively few mutations in this population. We believed this choice was conservative and constitutes a strong test of our hypothesis since fewer $\lambda$ mutations would provide fewer opportunities to detect host-induced contingency. We sampled $\lambda$ strains from different timepoints of population 'D7' and sequenced their *J* gene (*Figure 3a*, *Supplementary file*

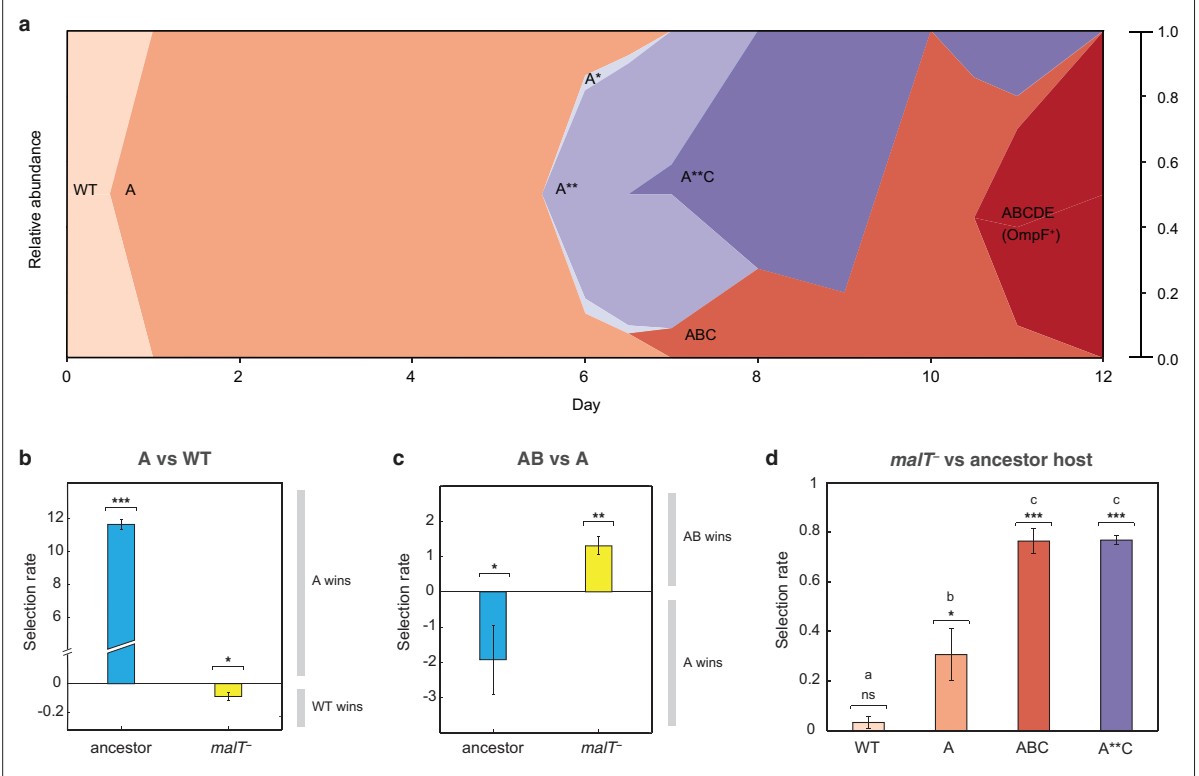

**Figure 3.** *J* evolution (**a**) and evidence of the interdependency between λ and *E. coli* fitness during their coevolution (**b-d**). (**a**) Phylogenetic reconstruction and relative abundance of λ genotypes isolated through time from a previously coevolved community (*Meyer et al., 2012*). Each letter and star indicate a non-synonymous mutation in *J* (see *Supplementary file 1c* for labels' corresponding mutations). A genotype's relative abundance on a given day is denoted by the fraction of the total height of the y-axis that it occupies (e.g. on day 9, frequency of ABC is 0.2 and A**C is 0.8; see *Supplementary file 1d*). The lineage WT-A-ABC-ABCDE eventually evolves OmpF function and fixes in the population; resistance in *E. coli* through *malT⁻* rises to high frequencies between days 5 and 8 (*Meyer et al., 2012*). (**b & c**) Selection rates (per 24 hr) of phage genotypes on the two hosts. Each bar represents the mean of three experimental replicates. While mutation A is favored over wildtype (WT) λ in the presence of the ancestral host and not *malT⁻*, AB only outcompetes A in the presence of *malT⁻* and not the ancestral host. One tailed *t*-tests to test if the mean selection rate is significantly greater (or less) than zero: A vs WT with ancestor host- $t = 98.76$, $d.f. = 2$, $p < 0.0001$; A vs WT with *malT⁻* - $t = -4.99$, $d.f. = 2$, $p = 0.0190$; AB vs A with ancestor- $t = 3.4$, $d.f. = 2$, $p = 0.0383$; AB vs A with *malT⁻* - $t = -8.88$, $d.f. = 2$, $p = 0.0062$. (**d**) Selection rate (per 4 hr) of *malT⁻ E. coli* relative to its ancestor in the presence of λ from different stages of coevolution. Each competition was replicated three times. Lowercase letters denote significance via Tukey's honest significance test, see *Supplementary file 1g* for pairwise p-values (ANOVA: F-ratio=111.22, *d.f.*=11, p<0.0001). One tailed *t*-tests were also used to test if the selection rate of *malT⁻* was greater than zero WT: $t = 2.44$, $d.f. = 2$, $p = 0.676$; A: $t = 5.12$, $d.f. = 2$, $p = 0.0181$; ABC: $t = 26.59$, $d.f. = 2$, $p = 0.0007$; A**C: $t = 71.67$, $d.f. = 2$, $p < 0.0001$. This shows that *malT⁻* is unlikely to evolve in the presence of WT λ but it becomes progressively more likely as λ gains mutations. Asterisks over all the competitions indicate significance level corresponding to the p-values. Error bars in all bar graphs represent one sample SD.

The online version of this article includes the following figure supplement(s) for figure 3:

**Figure supplement 1.** Competition assay of λ isolates from population D7.

*1c* and *Supplementary file 1d*). Next, we ran pairwise competition experiments between λ genotypes at different stages of evolution on the two hosts. We found that the first mutation on the line of descent to OmpF⁺ required ancestral *E. coli* to evolve, while the second mutation required *malT⁻ E. coli* (*Figure 3b and c*). In addition, the eventual OmpF⁺ genotype with five *J* mutations only outcompeted the genotype with two mutations when provided with *malT⁻* hosts (*Figure 3—figure supplement 1*). These findings show that the path λ took in population D7 required it to sequentially adapt to both host types and that λ's fitness landscape changed during coevolution in a way that ultimately facilitated evolutionary innovation.

At this point, we shifted the focus of our study to testing whether *E. coli*'s resistance evolution was also impacted by λ's evolution. Before reconstructing *J* evolution, it was believed that *E. coli* evolved resistance first and then *J* mutations evolved in response (*Meyer et al., 2012*, Figure 5). However, *J* evolved within a day, while *malT⁻* mutations were previously shown to fix between 5 and 8 days in the

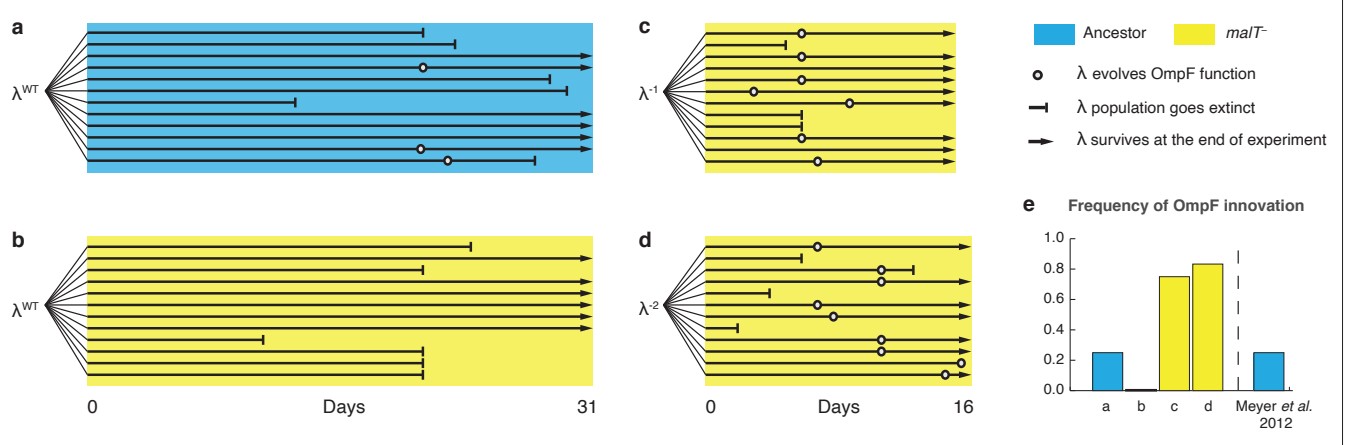

**Figure 4.** Evolutionary replay experiments reveal that λ's evolution to use OmpF depends on host coevolution. (**a**) Wildtype (WT) λ with ancestral host, (**b**) WT λ with *malT⁻*, (**c**) λ one mutation removed from evolving OmpF function with *malT⁻*, and (**d**) identical setup as (**c**), but with λ two mutations removed (see ***Supplementary file 1f*** for identity of mutations). (**e**) The bar graph provides the frequency of OmpF⁺ evolution compared to the frequency observed by ***Meyer et al., 2012***. Given λ's established one in four rates of OmpF⁺ with ancestral host, the probability of observing no OmpF⁺ evolution in 12 replicate populations is ~0.03. Thus, no positives for OmpF evolution in (**b**) shows that λ's evolution to OmpF function is significantly hindered, when coevolution is initiated with *malT⁻* host. However, *malT⁻* does not impede OmpF⁺ evolution, when the coevolution is initiated with already evolved λ s (p-values for Fisher's exact test: between (**b**) and (**c**) p=0.0261; between (**b**) and (**d**) p=0.0122). Notably, some λ populations went extinct which is common for these experiments and was previously shown to be caused by the evolution of resistance mutations in *E. coli*'s ManXYZ protein complex (***Meyer et al., 2012***).

The online version of this article includes the following figure supplement(s) for figure 4:

**Figure supplement 1.** A schematic overview of the coevolution experiments performed in this study and by ***Meyer et al., 2012***.

**Figure supplement 2.** Phage densities during the coevolutionary replay experiment performed in ***Figure 4***.

population (***Meyer et al., 2012***). The timing suggests that λ improved infectivity and then applied pressure on *E. coli* to evolve resistance. To test whether λ evolution promoted host resistance evolution, we ran competition experiments between ancestral and *malT⁻* hosts in the presence of phages isolated from four different time points. We found that *malT⁻* was not significantly more fit than the ancestral *E. coli* in the presence of the ancestral λ, but it was more fit in the presence of the evolved λ s (***Figure 3d***). This provides another example of interspecific epistasis, but this time the parasite's genotype alters the host's landscape. This result combined with the others suggests that there is an intricate coevolutionary feedback at play between λ and *E. coli*: λ evolves *J* mutations that better exploit *E. coli*, which in turn applies pressure on *E. coli* to evolve resistance. Once resistance evolves, new adaptive pathways become available to λ that encourage the innovation. For the computer simulations, we arbitrarily chose timepoints to switch from one host to the other; however, in reality, the dynamics of the switch are dictated by the host-parasite coevolution.

## Evolutionary replay experiments

To further test the role of coevolutionary processes at driving λ's innovation, we ran replays of the coevolution experiment (***Figure 4—figure supplement 2***). We initiated 12 populations with *malT⁻* host that already possessed resistance, and 12 populations with ancestral host where λ and *E. coli* would coevolve normally. The former treatment should hinder the evolution of OmpF function because it denies λ the opportunity to evolve first with ancestral *E. coli*. In line with our expectations, 0 of 12 replicates evolved OmpF use in the *malT⁻* treatment, meanwhile 3 out of 12 evolved the innovation in the ancestral treatment (***Figure 4a and b***). By sequencing *J* alleles of the resulting λ genotypes, we found that fewer mutations evolved with *malT⁻* despite evolving for the same length of time. This suggests that λ's evolution was stymied by starting with the resistant host (***Supplementary file 1e***), and by disrupting the coevolutionary process we interfered with λ's ability to innovate.

Lastly, we tested whether *malT⁻* would still promote the evolution of OmpF use if the replay experiments were initiated with λ genotypes positioned further along the path to gaining OmpF function. We initiated two more replays: one with a λ strain that was just a single mutation away from becoming

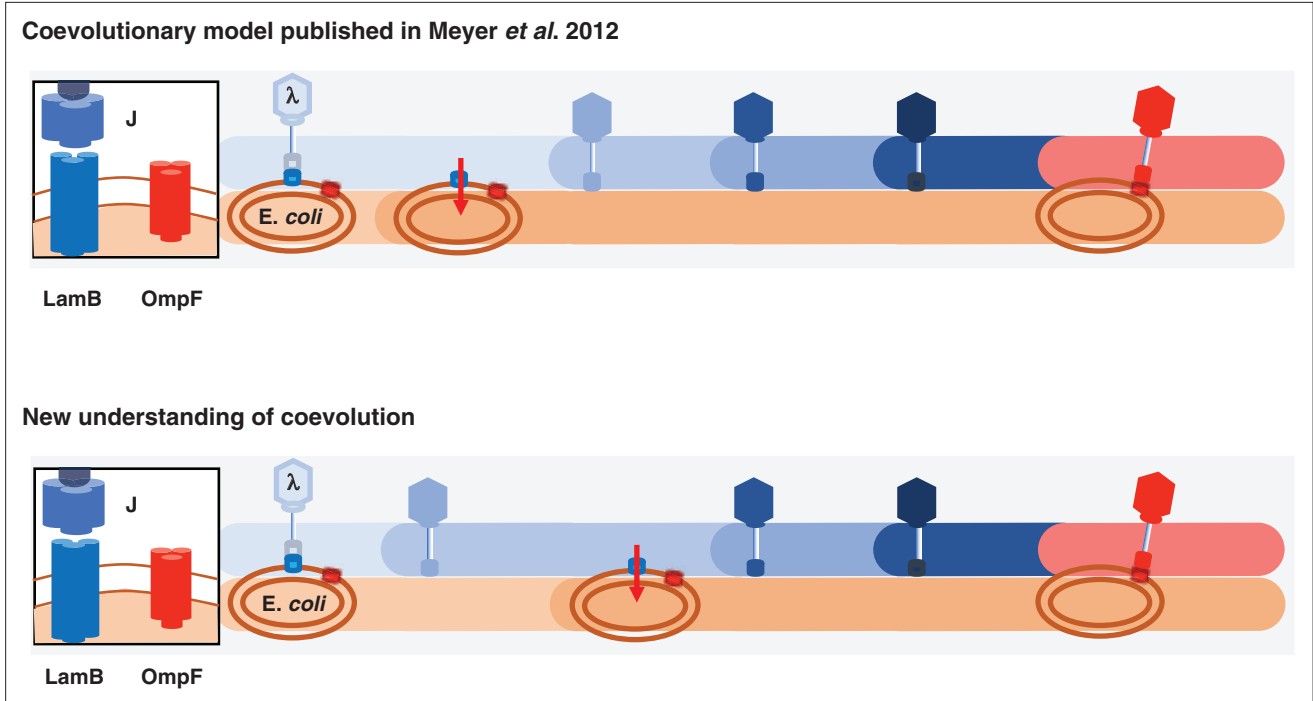

**Figure 5.** Previous and new version of $\lambda$-*E. coli* coevolutionary model. In the original model, *E. coli* evolves resistance by repressing $\lambda$'s receptor and then $\lambda$ evolves mutations that allow it to use a new receptor. The updated model describes a more involved dynamic where $\lambda$ evolves mutations that improve its ability to infect *E. coli*; with increased pressure to avoid infection, E. coli responds by evolving resistance, and then $\lambda$ evolves the remaining mutations that leads to the ability to infect using the new receptor.

OmpF⁺, and another that was two mutations away (see *Supplementary file 1f* for J alleles). 16 of 24 $\lambda$ populations evolved OmpF⁺ showing that whether *malT⁻* promotes OmpF⁺ evolution depends on where the $\lambda$ genotypes are located in the fitness landscape (*Figure 4c and d*).

While the goal of these experiments was to test a specific hypothesis about $\lambda$'s innovation, the results have broader implications for coevolutionary dynamics and the repeatability of evolution. We showed that if the host outpaces the parasite (*Figure 4b*; $\lambda$ WT and *malT⁻ E. coli*), then the parasite is unable to innovate. This is in line with a previous study showing that if the host evolves even higher levels of resistance than provided by the *malT* mutation, $\lambda$ loses the ability to innovate (*Meyer et al., 2012*). When specific pairs of host and parasite genotypes are combined that are at complementary steps in the coevolution, then the innovation becomes possible and even highly likely (*Figure 4a, c and d*; $\lambda$ WT and ancestral *E. coli*, $\lambda$⁻¹ and *malT⁻ E. coli*, and $\lambda$⁻² and *malT⁻ E. coli*). This suggests that the genotype of host and parasite that first encounter each other in nature, as well as the timing of their coevolution, play an important role in determining the dynamics and endpoints of their evolution.

## Discussion

This study provides multiple direct tests of coevolution's role in driving innovation, as well as revealing a more complicated model of $\lambda$ and *E. coli* coevolution than previously published (*Figure 5*). $\lambda$ was thought to evolve OmpF use as a direct response to *malT⁻* resistance (*Meyer et al., 2012*); however, here we learned that key steps were missing from that model. $\lambda$ is initially poor at infecting its host, so it evolves mutations in J that enhance its infectivity. The new $\lambda$ genotypes apply pressure on *E. coli* to evolve resistance. When host-resistance increases in the community, $\lambda$'s fitness landscape is deformed in a way that promotes J evolution toward OmpF use, but only if it had already acquired some J mutations. Remarkably, the timing and coordination of each of these interdependent steps is facilitated by the reciprocity ingrained in host-parasite coevolution. Altogether, we were able to provide direct experimental evidence that fluctuating landscapes, also known as fitness seascapes (*Merrell, 1994*), can promote evolutionary innovations.

Our studies show that the fitness of a parasite depends on complex genetic interactions within its own genome and with the genomes of interacting hosts. These interdependencies result in highly contingent evolution, where $\lambda$ is unlikely to evolve an innovation unless it participates in a particular sequence of coevolutionary steps with its host. Despite the stochasticity that is expected to arise in systems with substantial historical contingency (*Gould, 1989*; *Blount et al., 2018*), $\lambda$'s evolution to use a new receptor is repeatable because the sequence is coordinated by coevolutionary feedback. While coevolution may yield tangled banks of interactions, we demonstrate how high-throughput technologies can be used to untangle them and to predict evolution. The ability to successfully predict evolution in any system represents a significant step forward, but it is particularly notable in conditions that incorporate species interactions. Our approach was data-intensive and relied on technologies that are not currently available for more natural and complex ecological systems; however, many efforts are underway to develop these technologies (*Bergelson et al., 2021*). This study shows that these efforts are worthwhile because even though the resulting data may appear to be an uninterpretable morass, computational analyses can be leveraged to penetrate the information to aid learning and prediction.

## Ideas and speculation

This work was completed during the 2020–22 SARS-CoV-2 pandemic, raising the question of whether this research provides insight into strategies to prevent future pandemics. While it is difficult to extrapolate, there are general lessons that can be applied to understanding pandemics. First, natural coevolutionary processes can repeatedly drive the evolution of viruses to achieve difficult adaptations that underly host-range expansions. On one hand, this knowledge is alarming because it suggests that pandemics may be more likely than previously anticipated, but on the other hand, the repeatability provides an opportunity to study the process and develop intervention strategies. We also learned that shifting conditions can enhance the evolutionary potential of viruses, suggesting that there could be additional unexpected consequences of human-caused environmental changes.

# Materials and methods
## Bacterial and phage strains

The ancestral phage $\lambda$ strain in our study is cI26; it is a strictly lytic strain that was used in the previous coevolution study on which this paper builds (*Meyer et al., 2012*). $\lambda$ has two life cycles: lytic and lysogenic. In the lytic life cycle, $\lambda$ infects its host, creates multiple copies of its genome, and lyses the cell to produce new viral particles. This is unlike the lysogenic life cycle of $\lambda$, where it stably integrates its genome into the host genome and is replicated along with the host (*Hendrix, 1983*). The $\lambda$ strain cI26 has a frameshift mutation in the *cI* gene that disrupts the regulatory protein cI required by $\lambda$ to switch to a lysogenic life cycle (*Meyer et al., 2012*). This renders cI26 obligatory lytic. The other $\lambda$ strain we used was cI857 (provided by Ing-Nang Wang, State University of New York, Albany) to construct our $\lambda$ genomic library using MAGE (discussed in 'Fitness landscapes' section). The advantage of using cI857 over cI26 is that it is able to form a lysogen and as a genomic complex with *E. coli*'s genome, its genome can be easily edited with *E. coli* engineering methods. cI857 has an advantage over typical lysogenic $\lambda$ because it has a mutation in the *cI* gene that makes the repressor protein cI unfold at high temperatures (*Meyer et al., 2016*). This enables us to induce the lytic life cycle of cI857 from lysogens using heat shock treatments, which has fewer side effects compared to using mutagens that the typical strain requires for induction.

We used *E. coli* B strain REL606 for the ancestral-sensitive host and its derivative EcC4 for the evolved-resistant (*malT*⁻) host to construct fitness landscapes (in *Figure 1*) and run competition assays (in *Figure 3*). REL606 was the ancestral host in the coevolution experiments performed by *Meyer et al., 2012*, and EcC4 was isolated from one of the coevolving populations that has a single mutation, a nonsense mutation (C→T) at genome location 3,482,567 in *malT* gene (Table S5 in *Meyer et al., 2012*). MalT is a positive regulator of *lamB*, so a disruption in transcription of *malT* inhibits LamB expression (*Boos and Böhm, 2000*). Resistance to $\lambda$ in EcC4 due to this nonsense *malT* mutation has been shown to yield high levels of resistance, the equivalent of a nonsense mutation in *lamB* (Figure S5 in *Chaudhry et al., 2018*). We used another *malT*⁻ mutant of REL606, named LR01, for our coevolutionary replay experiments (*Figure 4*). This strain has a 25 bp duplication at genome location

3,482,677 causing a frameshift in *malT*. LR01 was also isolated from a coevolving population of $\lambda$ and *E. coli* (Figure S3 in *Meyer et al., 2012*); however, unlike EcC4, LR01 has a high reversion rate to *malT*⁺ that results in 'leaky resistance' to $\lambda$ (*Chaudhry et al., 2018*). This allows $\lambda$ populations to sustain serial dilution transfers and thus was critical for the success of the coevolutionary replay experiments (*Figure 4*). We note that while we use the term 'resistant' to refer to *malT*⁻ genotypes, this 'resistance' is specific to the OmpF⁻ $\lambda$ in this study; these bacterial strains are susceptible to OmpF⁺ $\lambda$ phage.

Several *E. coli* strains were used for culturing $\lambda$. The most often used was DH5α, a *lacZα*⁻ derivative of *E. coli* K-12, because it is permissive to all $\lambda$ genotypes and it lacks *lacZ* which is used as a genetic marker to distinguish phage genotypes in competition assays (additional information in the 'Phage Competition Experiments' section). Two *lamB*⁻ mutants of *E. coli* with non-functional LamB were used interchangeably to culture OmpF⁺ $\lambda$ on LB agar. One strain was a derivative of REL606 that has a 1 bp insertion of nucleotide T between base positions 610 and 611 in *lamB* (*Meyer et al., 2012*; *Meyer et al., 2015*), and the other strain was an *E. coli* K-12 derivative from the Keio collection (*lamB*⁻ JW3996) (*Baba et al., 2006*). We found no difference in the efficiency of plaquing on the two strains.

MAGE was performed in an *E. coli* K12 strain, HWEC106, provided by Harris Wang, Columbia University. The strain's *mutS* gene is deleted and it possesses the pKD46 plasmid with an inducible $\lambda$-red recombineering system (*Datsenko and Wanner, 2000*). cI857 successfully integrated into this strain's canonical ATTB site located near the *galK* gene (*Meyer et al., 2012*; *Maddamsetti et al., 2018*; *Petrie et al., 2018*).

## Media

We performed most experiments in conditions identical to the initial coevolution experiment (*Meyer et al., 2012*). For competition assays (low and high-throughput) and two of the four coevolutionary replay experiments (*Figure 4a and b*), bacteria and phage were cocultured in modified M9 Glucose (47.7 mM disodium phosphate, 22.0 mM potassium phosphate monobasic, 18.7 mM ammonium chloride, 8.6 mM sodium chloride, 0.1 mM calcium chloride, 10 mM magnesium sulfate, 5.55 mM glucose, and 0.2 mM of LB) as used in *Meyer et al., 2012*. A new medium was used for the remaining two coevolution replays (*Figure 4c and d*) that we call Tris DM Glucose: 1.6 mM potassium phosphate monobasic, 0.59 mM potassium phosphate dibasic, 0.2 mM of calcium chloride, 50 mM Tris base (pH 7.4), 10 mM of magnesium sulphate, 7.5 μM thiamine, 3.2 mM ammonium sulphate, and 5.55 mM glucose. This medium is improved over M9 Glucose because it is less prone to magnesium precipitation, and fortunately has no noticeable effect on the coevolution.

For isolation, estimating densities and initial culturing of bacteria and phage, four more media were used according to the following specifications—(a) LB (Lennox Broth): 10 g tryptone, 5 g yeast extract, and 5 g sodium chloride per liter of water, (b) LBM9: 20 g tryptone, 10 g yeast extract, 12.8 g sodium phosphate heptahydrate, 3 g potassium phosphate monobasic, 0.5 g sodium chloride, 1 g ammonium chloride, 1.2 g magnesium sulfate, 22 mg calcium chloride per liter of water, (c) LB agar: 10 g tryptone, 5 g yeast extract, 5 g sodium chloride, and 16 g agar per liter of water, (d) soft agar: 10 g tryptone, 1 g yeast extract, 8 g sodium chloride, 7 g agar, 0.1 g glucose per liter of water, supplemented with a final concentration of 2 mM calcium chloride. The soft agar was also supplemented with 10 mM magnesium sulphate to improve plaquing of phage particles.

## Isolation and culturing techniques

Most strains were grown at 37°C, except for HWEC106, which was grown at 30°C because pKD46 has a temperature-sensitive origin of replication. To ensure uniform aeration and nutrient availability, 4 ml cultures were shaken at 220 rpm and 10 ml cultures at 120 rpm; different rpm was chosen to ensure comparable aeration in cultures (10 ml cultures were grown in 50 ml Erlenmeyer flasks and 4 ml in glass tubes). We used LBM9 to grow phage cultures and LB to grow bacteria cultures, unless otherwise indicated. All bacteria and phage stocks were stored by freezing 1 ml of culture with 15% glycerol at −80°C. We revived phage from freezer stocks by growing ~2 μl of frozen stocks on 100 μl of DH5α overnight culture in 4 ml of LBM9. To harvest phage, cells were killed and separated by adding 100 μl of chloroform and centrifuging the solution at 3900 rpm for 10 min. The supernatant containing phage lysate was then stored with ~2% chloroform at 4°C. Bacteria were revived from −80°C by growing ~2 μl of the frozen stocks over night in 4 ml of LB.

Phage strains were isolated from a population by infusing phage particles into bacterial lawns of DH5α cells. We made these infused plates by mixing a small volume (between 10µl and 100 µl) of diluted phage and ~5 × 10^8 host cells to a 4 ml of molten soft agar at 55°C, and pouring the mixture over an LB agar plate (*Sambrook and Russell, 2001*). The soft agar was allowed to solidify and then incubated overnight at 37°C for phage particles to form plaques. A plaque is a near-circular clearing that forms on the bacterial lawn when nearby cells are killed by an infection that is initiated by a single phage particle. Phage dilutions were made in saline solution (8.5 g/L NaCl) and implemented to yield between 30 and 300 plaques. A single plaque was picked from the infused plates and clonal phage stocks were made by culturing the isolate overnight with DH5α and extracting phage lysate using techniques similar to frozen $\lambda$ stocks.

### Estimation of phage densities

Infused plates with appropriate dilutions of phage in saline solution (8.5 g/L NaCl) were used to estimate phage densities for phage competition assays. We controlled the number of plaques on an infused plate so that individual plaques could be identified and counted. These counts were used to back-calculate the phage growth rate.

To estimate the phage densities in the coevolutionary replay experiments, serial dilutions of phage were spotted on a bacterial lawn of DH5α. We made the bacterial lawns by adding ~5 × 10^9 cells of DH5α to 10 ml of molten soft agar at 55°C, and pouring it over a 150 mm diameter petri dish of LB agar base (*Sambrook and Russell, 2001*). After the soft agar solidified, 2 µl of eight different phage dilutions were added on top of the surface, let to dry, and incubated overnight at 37°C. The plaques were counted from the dilution where we could identify individual plaques. We then used these counts to calculate phage density in pfu per ml, where pfu is plaque forming units.

### Sequencing for analyses of mutations in *J*

We routinely sequenced the *J* gene of $\lambda$ to verify the identity of stocks and to determine whether *J* evolved in the replay experiments. DNA samples were PCR amplified using Q5 high-fidelity 2 × master mix (New England Biolabs) and primers described in *Supplementary file 1h*. Unpurified PCRs were submitted to Genewiz (La Jolla, CA) for Sanger sequencing.

### Calculation of selection rate using Malthusian parameter

Selection rate (*s*) was used to quantify the difference in fitness between pairs of strains (say X and Y) in a given environment. It is calculated as the difference of their Malthusian parameters:

$$s = \frac{ln\frac{X_T}{X_0} - ln\frac{Y_T}{Y_0}}{T} \qquad (1)$$

where $X_t$ and $Y_t$ are the densities of strains X and Y, respectively, at time $t$, and $T$ is the period of time over which strains are assessed (and the assay starts at $t = 0$). Selection rate has units of inverse time; however, in all our figures, we report selection rate as per unit time period ($T$) of the assays. We reported the values this way because $\lambda$ does not strictly grow exponentially and it can be misleading to report normalized rates after dividing by $T$ because the rates cannot be extrapolated for different times in a straightforward way like for exponential growths. That being said, for readers who want the per hour rates, we have provided the value of $T$ in hours within each figure caption.

### Construction of $\lambda$ genomic library using MAGE

Our goal was to measure the fitness of a significant number of $\lambda$ genotypes to establish the basic structure of the landscape. To accomplish this, we generated a combinatorial library of genotypes made with 10 *J* mutations previously observed to evolve en route to OmpF^+ (*Supplementary file 1a*). These mutations were chosen because they fell within two <100 base frames where adaptive mutations tended to evolve. We used a genetic engineering technique called MAGE to construct the library (*Wang et al., 2009*). This technique employs the $\lambda$-red recombineering system that can efficiently recombine single-stranded DNA into *E. coli*'s genomic DNA. To engineer $\lambda$, $\lambda$'s genome is integrated into *E. coli*'s genome, creating a lysogen. The $\lambda$ genome in the lysogen (a prophage) becomes dormant and can be treated as any other *E. coli* gene. Gene edits in this $\lambda$ lysogen are made by expressing $\lambda$-red from a plasmid (pKD46) within *E. coli* cells (*Datsenko and Wanner, 2000*) and

electroporating synthetic oligos with specific mutations written into the sequences. Multiple oligos can be combined into a single experiment in order to create a diverse library of different combinations of the mutations. The output of this procedure is a single sample with a library of genotypes mixed together.

We previously reported our MAGE protocol in *Maddamsetti et al., 2018*. The oligos used are in Table S1 of the manuscript listed under the subheading '10-mutation library'. There are a few important aspects of the oligo and protocol design that we discuss here, but for more detailed information see the original paper.

Our goal was to construct a genetic library with each combination of mutations represented equally, this means a 50/50 split of mutant and WT states at each site. To achieve this, we modified the typical MAGE protocol. First, when mutations were clustered near each other such that a single 90-mer used for MAGE editing encompassed multiple mutations, we designed multiple oligos with different combinations of the mutations at these nearby sites. Without doing this, edits would be correlated, creating an imbalance in the network representation, and limiting our ability to determine the effects of individual mutations. Second, we designed oligos with the WT state at the 10 positions. Including these oligos slowed the efficiency of MAGE to introduce mutations because sometimes they would overwrite an edit; however, it safeguarded the procedure from saturating the library with the 10 mutations. A third strategy was employed to even out the variation at these 10 sites. This was to perform MAGE starting from two orthogonal points in the genetic network; $\lambda$ that had none of the 10 mutations and one that we engineered in all 10 (*Maddamsetti et al., 2018*). Additionally, we ran MAGE on three separate replicates from each starting point in order to enhance the potential to explore more genetic space. Lastly, we ran the MAGE protocol for 50 cycles, which is the number of cycles a computer simulation of MAGE with an efficiency of 10% recombination efficiency per cycle predicted we would need to construct all variants. In the end, we mixed all six MAGE libraries together in equal frequencies to maximize diversity in the library.

Our design also included measures to improve our ability to detect rare variants in the library. The next step of the fitness landscape protocol is amplicon sequencing with an Illumina MiSeq. This technology has relatively low rates of error ($\sim 10^{-3}$ per base sequenced); however, we anticipated that some of the genotypes in our library would fall below this frequency, making it impossible to distinguish between false positives and true reads. To improve our resolution, we edited a synonymous mutation adjacent to each focal mutation. We called this a watermark mutation because it helped us distinguish between a true genome edit and an error in sequencing. By only recording the presence of a focal mutation if it occurred alongside the watermark, we decreased our detection limit from $\sim 10^{-3}$ to $\sim 10^{-6}$.

The watermark strategy we employed was slightly more complicated than this. Most synonymous mutations at the C-terminal end of proteins, like our watermark mutations, have no fitness effects (*Kelsic et al., 2016*). Despite this, we designed a way to test for neutrality. For each focal mutation, we edited one of two different watermark mutations (see *Supplementary file 1i* for list of mutations). This allowed us to compare equivalent genotypes with distinct watermarks in order to test whether one of the synonymous mutations impacted $\lambda$'s fitness when compared to the other. We found that there was no significant fitness effect of the neutral mutations on the phage genotype (see Appendix 1).

## Empirical fitness landscape resolved through one-pot competition experiment

After creating the $\lambda$ genomic library, we measured the relative fitness of genotypes with respect to the WT $\lambda$. To do this, we cocultured the $\lambda$ genomic library and ancestral $\lambda$ in a 1:9 ratio. This ratio was used so that the most abundant competitor remains the ancestor throughout the competition experiment. This means that the competitive fitness we measured is with respect to a single genotype. If we had not done this, then as the community of engineered $\lambda$ shifts during the competition, and more fit genotypes become enriched, they will change the mean fitness of the population and cause mildly fit genotypes to begin to decline, making them appear unfit. Flooding the flask with ancestral $\lambda$ and running competitions for a short time period (4 hr) solves this problem.

Having a disproportionate number of ancestral $\lambda$ has a pitfall. We measured $\lambda$ fitness by comparing the frequency changes of each genotype using amplicon sequencing (more information in the next section). The problem is that most of our sequencing effort would be spent on sequencing a single genotype, the ancestor. To avoid this, we engineered an ancestral genome with 6 synonymous

mutations, where the reverse primer binds. These edits interfered with primer binding and caused selective amplification of the library $\lambda$ s. We edited 6 'wobble' positions: 3381 (g→t), 3384 (c→a), 3387 (c→a), 3390 (c→t) 3393 (g→a), and 3396 (c→a). The engineering was done using MAGE with the oligo provided in *Supplementary file 1j*. These edits were made at the end of the protein (3399 nucleotide) and so they likely did not affect $\lambda$ fitness. We did not test this because this engineered strain acted as a standard competitor for all competition experiments and so it should have no effect on the relative fitness comparisons.

We ran eight competitions, four in the presence of ancestral *E. coli* and four with *malT⁻ E. coli*. Competitions were inoculated with ~$10^7$ total $\lambda$ particles and ~$2 \times 10^8$ cells into 10 ml of M9 glucose in 50 ml flasks. The cells were preconditioned in M9 glucose for 24 hr before the competition. Flasks were cultured for 4 hr at 37°C and shaking at 120 rpm. 1 ml samples were removed from the library before and after the competition for processing. Phage particles were concentrated using PEG precipitation (*Sambrook and Russell, 2001*). The pellet was resuspended in 25 µl of molecular grade water and then a two-step PCR reaction was performed in order to amplify just the region of interest, to attach barcodes to the amplicons so that we could multiplex samples into a single Illumina run, and attach adapters required for Illumina sequencing. The protocol was published by *Kelsic et al., 2016*. The primers we used are provided in *Supplementary file 1k*. As described in Kelsic et al., we ran two separate PCR reactions for each sample and used a unique barcode on each. This allowed us to test for amplification bias, which we did not detect (*Figure 1—figure supplement 2*).

## Sequencing

Amplicon samples were pooled together and sequenced using an Illumina Mi-Seq maintained in the Systems Biology Department at Harvard Medical School. 100 base paired-end reads were run. We were able to extract on average 107,839 high quality reads per sample with standard deviation of 26,889. This provided considerable coverage to reliably estimate fitness for even rare genotypes.

## Post-sequencing analysis and construction of fitness landscape

Amplicon sequences were analyzed to calculate selection rates for each engineered genotype. These values were later used to construct the fitness landscapes. We used a combination of custom Python and MATLAB (version R2019b) scripts to read and concatenate the raw paired-end reads, identify the genotypes based on the focal mutations they possessed, and to count their abundances. All focal mutations were called only if either of the two corresponding watermark mutations were also present. We removed any reads that contained more than one mutation other than the focal and watermark mutations. This essentially acted as a quality filter and we did not have to filter based on the Illumina provided Q-score. For each competition, a mean of counts corresponding to the two barcodes used in PCR amplifications was taken for all the genotypes. If the counts for both, initial and final timepoints were available for a genotype, they were used to estimate the genotype's Malthusian growth rate, given by the calculation $\frac{\ln \frac{\lambda^{i,T}}{\lambda^{i,0}}}{T}$ where $\lambda_{i,t}$ is the density of the given genotype at time $t$ and T is the total time of the competition. Note that most genotypes performed poorly on *malT⁻* host and fell below detection limit at the final timepoint. We quantified the differences in fitness of all genotypes by calculating their selection rate with respect to the WT $\lambda$ (see 'Calculation of selection rate using Malthusian parameter' section). The final fitness landscapes (*Figure 1a and b*) were constructed by taking the mean of the selection rates obtained from the four replicate competition trials performed on the same host.

## Statistical analysis of fitness landscapes

We used multiple linear regression models to quantify the genetic interactions in the fitness landscapes. We regressed fitness values on each landscape against individual effects of the mutations and the pairwise epistatic interactions between them (genotype-by-genotype or G×G interactions). This allowed us to estimate the contribution of epistasis toward explaining the observed fitness data. The fitness of each genotype in the regression model was thus described by 55 predictor variables (10 main effect terms for the 10 mutations +45 interaction terms):

$$y = \beta_{**} + \sum_i \beta_{i*} \, G_i + \sum_{i<j} \beta_{ij} G_i G_j \qquad (2)$$

where $\beta_{**}$ is the intercept, all other βs are regression coefficients, contribution of an individual mutation $i$ is described by the term $G_i$ , and the effect of pair of mutations $i$ and $j$ are captured by terms $G_iG_j$ , where $G_i$ is an indicator variable that is equal to zero when the mutation $i$ is absent and one when present. After fitting the model, we used Benjamini-Hochberg procedure (**Benjamini and Hochberg, 1995**) to control for false discovery rates and identify statistically significant terms in the model (see **Figure 1c**). All analyses were performed in R version 3.6.1 (**R Development Core Team, 2019**). The error in fitness estimates of the genotypes did not depend on the mean fitness value of the genotypes (**Figure 1—figure supplement 4**).

To understand how $\lambda$'s landscape differed between the two hosts, we incorporated the host genotype as a predictor variable in the linear model and regressed the combined fitness data of both landscapes together. This allowed us to test if a mutation significantly interacted with the host (genotype-by-environment or G×E interaction) and whether interaction between pairs of mutations changed with the host genotype (genotype-by-genotype-by-environment or G×G×E interaction). The combined full-factorial model consisted of a total of 111 terms (10 G for individual mutation +1 E for host +45 G×G+10 G×E+45 G×G×E terms),

$$y = \beta_{***} + \sum_i \beta_{i**}G_i + \beta_{**+}E + \sum_i \beta_{i*+}G_iE + \sum_{i<j}\beta_{ij*}G_iG_j + \sum_{i<j}\beta_{ij+}G_iG_jE \tag{3}$$

where additional $E$ is an indicator variable for host type and other notations follow same scheme as in **Equation 2**. Since the fit of a model generally improves as the number of predictor variables increases, we tested for overfitting using Akaike information criterion (AIC) (**Akaike, 1974**). AIC is a penalized-likelihood criterion which penalizes a model for increasing number of parameters in it. The model with the greatest relative likelihood is considered to be the one with minimum AIC value. We minimized AIC value using the *step* function in **R Development Core Team, 2019** to uncover subsets of predictor variables that have high predictive power. This resulted in a more parsimonious model (77 terms out of a total 111, see **Figure 1d**). A complimentary adjusted R-squared analysis came to the same conclusion, $R^2_{adj} = 0.775$ for the reduced model and $R^2_{adj} = 0.769$ for the full-factorial model (higher values indicate more predictive and parsimonious models). After model selection, we used Benjamini-Hochberg procedure to determine which variables were significantly predictive (**Figure 1d**). This procedure only controls for false positives and is susceptible to false negatives. In this way, it is conservative for our purposes.

## Simulation of $\lambda$'s evolution on fitness landscapes

To test if the changes in $\lambda$'s fitness landscape facilitated its evolution to infect via OmpF, we simulated $\lambda$'s evolution on the two landscapes we measured and recorded whether OmpF+ genotypes arose to a high enough frequency that we would have detected them in the original laboratory evolution. We initiated $\lambda$ populations with no mutations and allowed them to evolve and mutate at any of the 10 focal sites used to construct the landscapes (**Supplementary file 1a**). For each treatment (**Figure 2**), we simulated 300 separate trials using a modified Wright-Fisher model with discrete generations and a fixed population size.

For each simulation, a new fitness landscape was constructed by assigning fitness values to all the genotypes ($2^{10}$=1024 genotypes). This was done in a way to account for error in estimating fitness and the error associated with imputing the values of missing data points. For the genotypes that had empirical fitness data available (**Figure 1a and b**), we did not simply average the values of the replicate fitness measurements but instead we performed a bootstrapping protocol in order to account for the error in the fitness estimate. We randomly resampled from the four replicate measurements (with replacement) and then computed the mean of the four. Some genotypes were not present in all four replicates, in these cases we resampled as many times as there were replicates. Genotypes with only one replicate data point were, thus, assigned the same corresponding fitness values in all the runs of the simulations. Next, we imputed fitness of the genotypes that were missing from the empirical landscape. A genotype with a missing fitness value was randomly chosen and assigned the mean of the fitness values of its nearest neighbors (one-mutation away genotypes) present in the landscape. This was iterated until the full genotypic space was complete. Note that the order in which genotypes are chosen can affect the value that is estimated for a given genotype. This is because as the landscape is filled in, each genotype will have more neighbors to draw values from. This means

that if a genotype is randomly chosen early, its fitness will be based on fewer neighbors than if it were chosen later, and its value will be slightly different. Since missing genotypes are randomly chosen in each iteration, the order will vary from one simulation to the next. This method introduces an extra source of variation in the simulation runs and captures uncertainty associated with the imputation of fitness values.

After constructing a complete fitness landscape, we evolved a $\lambda$ population through repeated cycles of reproduction, selection, and mutation. For each generation, reproduction in the population was simulated by a multinomial sampling where the number of trials was equal to the population size $N$ (set to ~$6.3 \times 10^9$ based on *Figure 4—figure supplement 2*) and the success probability associated with a genotype $i$ was given by $p_i = n_i w_i / \sum_i n_i w_i$ , where $n_i$ is the abundance of the genotype $i$, and $w_i$ is defined as the exponential of selection rate used in fitness landscape. Thus, the probability of $(k_1, k_2, k_3, \ldots k_m)$ offsprings for genotypes $1, 2, 3, \ldots, m$ would be:

$$ \binom{N}{k_1 \; k_2 \; k_3 \ldots k_m} \prod_{i=1}^{m} \left( \frac{n_i w_i}{\sum_j n_j w_j} \right)^{k_i} $$

To incorporate mutations, all genotypes whose frequencies increased were mutated as per $\lambda$'s mutation rate ($7.7 \times 10^{-8}$ per base per replication *Drake, 1991*). Consider a genotype (say $i$) that increased in abundance and let the number of additional individuals produced by this genotype be denoted by $z_i$ (with $z_i = k_i - n_i$). Each of these $z_i$ individuals retained its parent's genotype with a probability $e^{-\mu}$ (assuming a Poisson distribution for mutations). Otherwise, the individual was assigned to a random neighboring (i.e. mutant) genotype. Given $\lambda$'s mutation rate, the probability of multiple mutations is very small and was ignored here. We simulated this modified Wright-Fisher cycle of reproduction and mutation for 960 generations. It is difficult to know how many generations phages undergo because the evolved phages have a high spontaneous death rate (*Petrie et al., 2018*) and can also experience other sources of mortality. Given this, we decided to run the simulation for a somewhat arbitrary amount of time, 960 generations which corresponds to two doublings per hour for the 20-day experiment we are trying to replicate *Meyer et al., 2012*. This is likely an overestimate, which is unintuitively conservative for our purposes because more cycles will cause additional evolution and exploration of the fitness landscape, enhancing the possibility of OmpF⁺ evolution in the negative controls, and reducing our ability to detect treatment differences.

We evaluated whether $\lambda$ evolved OmpF⁺ by examining the population for genotypes that had the necessary mutations to be OmpF⁺. Previous studies revealed that OmpF genotypes must have four mutations (*Maddamsetti et al., 2018*). They all possess two specific changes (A3034G) and (G3319A) and a third change that can occur at positions 3320 or 3321. For the 10 mutations we studied, T3321A is the only third mutation that satisfies this requirement, meaning all OmpF genotypes must have three specific changes (A3034G, G3319A, and T3321A). Many $J$ mutations satisfy the last requirement (*Maddamsetti et al., 2018*), so we implemented what we call the '3+1' rule where genotypes are designated as OmpF⁺ if they have the three necessary $J$ mutations plus any additional mutation (see *Supplementary file 1i*). If any such genotype crossed the threshold of 5000 $\lambda$ particles during the course of a simulation, the $\lambda$ population in that run was marked to have evolved OmpF-function. We based the threshold value on the detection limit of OmpF⁺ genotypes in the original laboratory coevolution experiments (~500 pfu/ml, see 'Coevolutionary Replay Experiments' section).

We implemented host switching by first evolving $\lambda$ on the landscape measured with ancestral *E. coli*, stopping the simulation early, and quantifying the frequency of each $\lambda$ genotype. Next, we would initiate evolution on the *malT⁻* for the remainder of the time but starting with genotypes at the frequencies recorded on the ancestral landscape. A total of 11 different coevolution treatments were run by varying how long $\lambda$ evolved on the ancestral host landscape and *malT⁻* landscape (*Figure 2*). 300 simulation trials were run for each treatment. We calculated error in the simulations in order to detect significant treatment differences by batching the runs into 30 and estimating the frequency of OmpF⁺ for each subset. This allowed us to calculate a 95% confidence interval using a Student's $t$ distribution for the frequency of OmpF⁺ evolution in each treatment, and then to use an ANOVA coupled with Tukey's multiple comparison test to test for treatment differences.

## Phage competition assays

To test whether $\lambda$'s path to innovation requires sequential adaptation to host genotypes isolated from different stages of their coevolution, we studied the dynamics of $\lambda$ evolution in much more depth than previously reported. We focused on a single experimental replicate reported on in *Meyer et al., 2012*; 'D7' that evolved to be OmpF⁺ on the 12th day. Daily samples of the population were preserved, so we were able to revive phages from the full time series. We isolated 66 phages in total (*Supplementary file 1c and d*) and sequenced the full-length *J* gene from each in order to reconstruct the evolutionary dynamics (*Figure 3a*).

Next, we ran head-to-head competition experiments from key genotypes along the path to OmpF⁺, the WT and $\lambda$ with a single *J* mutation (A), double (AB), and quintuple (ABCDE) (*Figure 3b and c*, *Figure 3—figure supplement 1*). Our first step was to mark key strains with a gene that caused the plaques to be visually distinguishable from unmarked phage. The gene is *lacZα*, which in the presence of a 5-bromo-4-chloro-3-indolyl-b-D-galactopyranoside (X-gal), Isopropyl-β-D-thiogalactoside (IPTG), and a host cell like DH5α that lacks *lacZα*, produces blue plaques. To incorporate *lacZα* into $\lambda$'s genome, we used a gene fusion of *lacZα* with $\lambda$'s *R* gene (*Burmeister et al., 2016*; *Shao and Wang, 2008*). The fusion readily recombines into $\lambda$'s genome by the phage's endogenous recombination system, $\lambda$-red (*Ellis et al., 2001*). The procedure is straightforward, infect an *E. coli* strain that has a plasmid with the lacZ fusion (strain: SYP042; plasmid: pSwtRlaczalphaZalpha +RZ xl1 Blue Amp Blue provided by Ing-Nang Wang, Albany, NY) and some fraction of the $\lambda$ produced will have recombined with the plasmid. The recombinants are isolated by picking blue plaques. WT and AB were each marked. The marker has been shown to have a slight fitness effect; however, this would not significantly influence our measurements of fitness since we observed such large differences between strains (*Burmeister et al., 2016*; *Shao and Wang, 2008*).

We competed three pairs of phages, WT$_{lacZ}$ vs A, A vs AB$_{lacZ}$, and AB$_{lacZ}$ vs ABCDE, under two conditions: with ancestral *E. coli* or *malT⁻*. Competitions were run for a single 24 hr period under identical conditions in which the phage evolved (*Meyer et al., 2012*). Initial $\lambda$ density was between $10^4$ and $10^5$ particles per ml and the relative frequency of the strains was sometimes skewed in order to start with more of the unfit genotype. Fitness differences were so large that the less fit genotype would be overwhelmed and its frequency undetectable if they did not start with a numerical advantage. ~5 × $10^6$ exponentially growing cells were inoculated to each flask (5 × $10^5$ per ml). The cells were preconditioned by growing them overnight in the competition medium. Three replicate competitions were run for each treatment. Phage densities of the unmarked and marked $\lambda$ were determined by plating in typical soft agar plates, where we added 0.5 mg/ml of X-gal and 0.25 mg/ml IPTG to the molten soft agar. Densities were measured at t=0 and t=24 hr.

## Host competition assay

This competition experiment tested the role of $\lambda$'s evolution plays in promoting the evolution of host resistance. We competed the ancestral host (REL606) against the evolved-resistant host (EcC4) in the presence of different $\lambda$ genotypes that had increasing numbers of *J* mutations (*Figure 3*; WT (cI26), A (1-mutation), ABC (3-mutation), and A**C (4-mutation)). We initiated three replicate populations for each phage treatment. The competition was performed with identical conditions as the original evolution experiment and run for 4 hr to prevent $\lambda$ from evolving during the competition assay. Cells were preconditioned in modified M9 glucose for 24 hr before the competition. As with $\lambda$, fitness was measured by observing the change in frequency of the competitors over time. Frequencies were determined by plating a subsample of the populations onto tetrazolium maltose (TMal) agar plates (*Shuman and Silhavy, 2003*). REL606 produces white colonies on the plates, while EcC4 produces smaller red colonies. Before plating, the phages were removed by centrifugation of the cells and then removing the supernatant that contained the phage. Cells were resuspended in saline solution and the centrifugation and resuspension were repeated once more.

## Coevolutionary replay experiments

Would $\lambda$'s evolution to OmpF innovation proceed without initially evolving on an ancestral host? We tested this by replaying $\lambda$-*E. coli* coevolution but starting out with a host that already evolved resistance via a *malT⁻* mutation (strain LR01). We initiated 12 replicate populations with *malT⁻*, and another 12 with REL606 as a positive control. The replay experiment was run nearly identically to the

coevolution experiment performed by *Meyer et al., 2012* with cl26 as the ancestor phage, and daily sampling was done to detect the presence of OmpF$^+$ phage by spotting the phage on the bacterial lawn of *lamB$^-$* cells (detection limit ~500 pfu per ml) (*Figure 4—figure supplement 1*). The two differences were that the study was run longer (31 days) and daily estimates of $\lambda$ populations were made (*Figure 4—figure supplement 2*). Twelve replicates were chosen in order to have enough statistical power to determine whether or not a treatment reduces the chances of OmpF$^+$ evolution. In the previously reported study, only a quarter of populations (24 out of 96) evolved to use OmpF (*Meyer et al., 2012*); this result has also been replicated in our lab on two separate instances where 3 out of 12 populations gained OmpF-function. Assuming a binomial distribution and a true success rate of 0.25 for $\lambda$'s OmpF$^+$ evolution with the ancestral host, the probability of observing no replicate evolving to be OmpF$^+$ in 12 replicate populations is 0.03167. Thus, we can conclude that *malT$^-$* treatment reduced $\lambda$'s ability to evolve the innovation as compared to when coevolution is initiated with ancestral host. Two isolates from each population were sampled on day 26, and the reactive region of their *J* genes were sequenced using the Sanger sequencing method.

We ran two additional replay experiments as a positive control for any unintended side effects of initiating an experiment with *malT$^-$* host (LR01). A prediction from the fitness landscape simulation results was that if we started a replay with *malT$^-$* and a phage isolated from a later-stage of the coevolution, then the phage should evolve OmpF use. To test this, we genetically modified an OmpF$^+$ $\lambda$ by removing one mutation required for OmpF use ($\lambda^{-1}$), and then a second ($\lambda^{-2}$). The OmpF$^+$ genotype was a lysogen (cl857) that we had previously edited in seven *J* mutations (*Supplementary file 1f*), which is reported about in *Petrie et al., 2018*. The two new edits were made using MAGE and the oligos are reported in *Supplementary file 1j*.

The replay experiments were run identically, except because of an oversight, these replays were run in a slightly different minimal glucose medium called Tris DM (compared to the original coevolution experiment run by *Meyer et al., 2012*). The key difference in the medium is the buffer, it is a Tris buffer, not a phosphate buffer. Both mediums impose carbon limitation and have the same concentration of the single carbon source, glucose, and so population and evolutionary dynamics should not have been affected. We confirmed in an additional replay experiment that the medium does not affect the timing or repeatability of OmpF$^+$ evolution.

## Data and code availability

Amplicon sequencing data used to generate fitness landscapes in this study have been deposited in the NCBI Sequence Read Archive under BioProject PRJNA646809. The code and data that support the findings of this study are available at https://github.com/anigupta12/fitness-landscape-paper, copy archived at swh:1:rev:19292491a2768444bd009da8a3de6d25ce8ab18c (*Gupta, 2021*).

Sharing of biological material requires a material transfer agreement, described at http://blink.ucsd.edu/research/conducting-research/mta/index.html.

## Acknowledgements

We thank the James McDonnel Foundation and the NSF-DEB (1934515) for financial support and The Max Planck Society for supporting JG. We thank A Agarwal for help with the statistical analyses of fitness landscapes.

## Additional information

### Funding

| Funder | Grant reference number | Author |
| --- | --- | --- |
| National Science Foundation | 1934515 | Justin R Meyer Animesh Gupta |
| McDonnell Foundation | | Justin R Meyer |
| Max Planck Foundation | | Jenna Gallie |

| Funder | Grant reference number | Author |
| --- | --- | --- |

The funders had no role in study design, data collection and interpretation, or the decision to submit the work for publication.

## Author contributions

Animesh Gupta, Conceptualization, Data curation, Formal analysis, Investigation, Methodology, Project administration, Resources, Software, Validation, Visualization, Writing - original draft, Writing - review and editing; Luis Zaman, Conceptualization, Formal analysis, Methodology, Resources, Software, Validation, Writing - review and editing; Hannah M Strobel, Data curation, Investigation, Methodology, Resources, Writing - review and editing; Jenna Gallie, Alita R Burmeister, Conceptualization, Data curation, Investigation, Methodology, Resources, Writing - review and editing; Benjamin Kerr, Conceptualization, Formal analysis, Funding acquisition, Methodology, Resources, Supervision, Writing - review and editing; Einat S Tamar, Methodology, Resources, Validation, Writing - review and editing; Roy Kishony, Conceptualization, Funding acquisition, Methodology, Resources, Supervision, Writing - review and editing; Justin R Meyer, Conceptualization, Data curation, Formal analysis, Funding acquisition, Investigation, Methodology, Project administration, Resources, Supervision, Validation, Writing - original draft, Writing - review and editing

## Author ORCIDs

Animesh Gupta http://orcid.org/0000-0003-4374-335X
Jenna Gallie http://orcid.org/0000-0003-2918-0925
Einat S Tamar http://orcid.org/0000-0001-8693-7176
Justin R Meyer http://orcid.org/0000-0001-5566-8452

## Decision letter and Author response

Decision letter https://doi.org/10.7554/eLife.76162.sa1
Author response https://doi.org/10.7554/eLife.76162.sa2

# Additional files

## Supplementary files

- Supplementary file 1. All the supplementary tables.
- Transparent reporting form

## Data availability

Amplicon sequencing data used to generate fitness landscapes in this study have been deposited in the NCBI Sequence Read Archive under BioProject PRJNA646809. Other data that support the findings of this study is deposited at Dryad at https://doi.org/10.5061/dryad.ht76hdrhq and all code is made available at GitHub at https://github.com/anigupta12/fitness-landscape-paper, (copy archived at swh:1:rev:19292491a2768444bd009da8a3de6d25ce8ab18c).

The following datasets were generated:

| Author(s) | Year | Dataset title | Dataset URL | Database and Identifier |
| --- | --- | --- | --- | --- |
| Gupta A, Zaman L, Strobel HM, Gallie J, Burmeister AR, Kerr B, Tamar ES, Kishony R, Meyer JR | 2022 | Host-parasite coevolution promotes innovation through deformations in fitness landscapes | https://doi.org/ 10.5061/dryad. ht76hdrhq | Dryad Digital Repository, 10.5061/dryad.ht76hdrhq |
| Gupta A | 2020 | Fitness landscape of bacteriophage lambda with ancestral host and resitant host | https://www.ncbi.nlm. nih.gov/bioproject/ PRJNA646809 | NCBI BioProject, PRJNA646809 |

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

## Appendix 1

### Fitness effect of synonymous mutations used as watermarks in MAGE

We designed two watermark mutations for each focal mutation. Watermark mutations are synonymous mutations that fall within a few nucleotides of the focal mutation. These edits improved our ability to detect the presence of focal mutations within the MAGE library (*Supplementary file 1i*). In principle, these watermark mutations should not affect the fitness of $\lambda$ because the introduced mutations do not change the amino acids. However, we designed two different watermarks in order to test whether or not they influence $\lambda$ fitness. To test for fitness effects of the synonymous mutations, we evaluated the fitness of genotypes with just a single focal mutation, and then we split the count data from the competition experiments into two, one for watermark '1' and a second pool for '2'. We calculated the fitness of each group and compared their means using a *t*-test. The analysis was limited to just fitness measurements made for the treatment with the ancestral host because the single mutants did poorly on *malT⁻* host and many dropped below our limit of detection during the competitions. We were also only able to run the analysis on 7 out of 10 mutations because in three cases only one watermark was represented in the final counts data. We found that the fitness did not depend on the neutral marker for all seven using a Bonferroni corrected alpha value of 0.0071 (*Supplementary file 1l*). However, one of the seven was significant based on the uncorrected alpha value of 0.05. The effect on the selection rate was estimated to be 0.9947 with 95% confidence interval of 0.4758–1.5136. This value falls well within the error associated with fitness estimates (*Figure 1—figure supplement 4*). Because of our uncertainty of whether there was an effect of this synonymous mutation, and because the effect falls within normal levels of error, we did not take steps to correct for this possible effect.

## Appendix 2

### Simulation results when only using genotypes present in both the ancestral and *malT⁻* fitness landscapes

One possible complication with the fitness landscape analyses is that the *malT⁻* landscape is based on a subset of the ancestral landscape (131 vs 580 out of 1,024 possible genotypes). This could lead to potentially spurious comparisons of evolution on the two hosts. To control for this, we re-ran the simulations starting with only the genotypes that were present in both the landscapes. While the frequencies of OmpF⁺ populations did shift with this new analysis, the main result that switching landscapes enhances the frequency of OmpF⁺ evolution remained statistically significant (*Figure 2—figure supplement 2d*). The increased frequency in ancestor-only treatment stems from the increased stochasticity in estimating the landscape since more genotypes are imputed in panel d than a-c. This stochasticity leads to more OmpF⁺ evolution because the increased randomness in landscape formation increases the chances of producing viable pathways to OmpF use. Interestingly, the ancestral landscape OmpF⁺ frequency is equivalent to *malT⁻* in panel d. This similarity suggests that the frequency of OmpF evolution for *malT⁻* may be artificially high, which would mean that in reality there may be greater differences between this treatment and the fluctuating treatments.

### Simulation results when shifting is done between landscapes of the same host

Each time we impute fitness values of missing genotypes in a simulation run, a slightly different landscape is produced. This procedure raises the question of whether the increased frequency of OmpF⁺ in the shifting landscape simulations is due to structural differences between the two landscapes, or the random differences created by the imputation technique. To test this, we repeated simulations as previously discussed; however, this time we shifted to a newly generated landscape created from the same host-landscape data. OmpF⁺ did *not* evolve when the shift was made between two ancestral landscapes. $\lambda$ did evolve OmpF function in some replicates when the switching was made between two *malT⁻* landscapes; however, the frequency was significantly less than when the shift was made from ancestor to *malT⁻* (*Figure 2—figure supplement 2e and f*). Lastly, to control for the sparser landscape of *malT⁻* compared with the ancestor, we shifted between two ancestral host landscapes that were generated using only fitness values of genotypes present in both landscapes. We found the same qualitative result as for the *malT⁻* to *malT⁻* landscape shift (*Figure 2—figure supplement 2g and h*). These results show that the structural differences between the two host landscapes encourage OmpF⁺ evolution above and beyond what results from the noise associated with the imputation procedure.

