## [Editor Report]

This study uses the parlance and framing of the fitness landscape to articulate a co-evolution story between host and parasite. It utilizes a tractable system, bacteriophage λ and *E. coli*, to ask questions that unite different pillars of evolutionary theory – evolutionary genetics (via the fitness landscape analogy), co-evolution, and host-parasite interactions. The findings will be relevant to a number of audiences, and will likely spawn downstream studies that further interrogate the molecular specifics that underlie host-parasite co-evolution.

---

## [Decision Letter]

**Decision letter after peer review:**

Thank you for submitting your article "Host-parasite coevolution promotes innovation through deformations in fitness landscapes" for consideration by *eLife*. Your article has been reviewed by 2 peer reviewers, and the evaluation has been overseen by a Reviewing Editor and Aleksandra Walczak as the Senior Editor. The following individuals involved in the review of your submission have agreed to reveal their identity: Nkrumah Grant (Reviewer #1), Jeremy Barr (Reviewer #2).

Essential revisions:

1) Please pay close attention to comments from reviewers with respect to the choice of study sub-population (why did the authors choose population D7 from Meyer et al. 2012 )?

2) Other stylistic and communication-based considerations as described by the reviewers. Some involve figure clarity. Also, please consider comments made with respect to items that are in the supplemental material.

*Reviewer #1 (Recommendations for the authors):*

1. Overall the manuscript is written well, but it could use some soft editing.

2. Consider making the color scheme of the simulations and biological data different. For example, figure 2 shows simulation data in blue and yellow, whereas biological data is shown in figures, 3 (panels B and C) and 4.

3. Please clarify why you chose population D7 from Meyer et al. 2012 to analyze further. Why only one population? How generalizable are your results?

4. It would be extremely beneficial to readers if the figures describing your methods were expanded upon (add figure showing MAGE-seq and input/outputs of simulations).

*Reviewer #2 (Recommendations for the authors):*

• I recommend including and potentially reworking supplemental figure 1 into the main text to provide a visual overview of the experimental system and also encourage the authors to visually recreate the λ co-evolutionary mechanisms explained in text in lines 41-47 on pg2. This would help with the readability of the manuscript, especially for readers not familiar with this model system.

• Similarly, the abstract appears intentionally vague and perhaps too broad at times. The authors could attempt to rework this to more clearly explain the co-evolutionary mechanisms that were seen in the study.

• I don't understand the rationale for numbering the supplemental figures in the way the authors have done so. Having sub-supplemental figures for each main figure is not logical and I recommend simply labeling supplemental figures in order of appearance in the main text.

• I really enjoyed the experimental design and approach to disentangling these co-evolutionary dynamics. This is more a comment rather than something to be addressed, but the phylogenetic reconstruction of λ genotypes in Figure 3A shows the emergence of A**C and ABC genotypes from a similar genetic background. How likely do the authors think that the C genotype emergence twice simultaneously and have they instead considered recombination playing a role?

---

## [Author Response]

Essential revisions:1) Please pay close attention to comments from reviewers with respect to the choice of study sub-population (why did the authors choose population D7 from Meyer et al. 2012 )?

We have added the sentences: ‘Population ‘D7’ was chosen because λ evolved relatively few mutations in this population. We believed this choice was conservative and constitutes a strong test of our hypothesis since fewer λ mutations would provide fewer opportunities to detect host-induced contingency.’

2) Other stylistic and communication-based considerations as described by the reviewers. Some involve figure clarity. Also, please consider comments made with respect to items that are in the supplemental material.

We have followed the advice of the reviewers and made many stylistic changes. These are described in our response to the reviewers.

Reviewer #1 (Recommendations for the authors):1. Overall the manuscript is written well, but it could use some soft editing.

We thank the reviewer for their comment and suggestion. We have re-read the entire manuscript and made minor edits at 17 different places. None of these changes altered the content of the manuscript.

2. Consider making the color scheme of the simulations and biological data different. For example, figure 2 shows simulation data in blue and yellow, whereas biological data is shown in figures, 3 (panels B and C) and 4.

We carefully considered this suggestion; however, we chose to leave the color scheme as-is. The color scheme contrasts the two hosts and we wanted to keep it consistent throughout the manuscript. We have modified the y-axis label in figure 2 to make it clearer that the results are from computer simulations by adding ‘(*in silico*)’. We also change the figure caption’s first sentence to emphasize that the results stem from simulations. ‘Simulation results of the frequency of OmpF-use evolution observed when fitness landscapes were shifted at different frequencies.’

3. Please clarify why you chose population D7 from Meyer et al. 2012 to analyze further. Why only one population? How generalizable are your results?

We have added the sentences: ‘D7 was chosen because λ evolved relatively few mutations in this population. We believed this choice was conservative and constitutes a strong test of our hypothesis since fewer λ mutations would provide fewer opportunities to detect host-induced contingency.’

These results are generalizable given that they align with the fitness landscape simulations where we used J mutations evolved in tens of independent λ populations. They were also in line with results from the replay experiments where replicate λ populations were free to evolve any mutations – not just the ones evolved in D7 – and they showed no capacity to innovate when cocultured with *malT*^–^ hosts.

4. It would be extremely beneficial to readers if the figures describing your methods were expanded upon (add figure showing MAGE-seq and input/outputs of simulations).

Thank you for this recommendation. We have added a new figure (Figure 2—figure supplement 1) to describe the simulations and we have edited the figure supplement 1 of Figure 1 to provide a more detailed schematic illustration of MAGE-Seq to the reader.

Reviewer #2 (Recommendations for the authors):• I recommend including and potentially reworking supplemental figure 1 into the main text to provide a visual overview of the experimental system and also encourage the authors to visually recreate the λ co-evolutionary mechanisms explained in text in lines 41-47 on pg2. This would help with the readability of the manuscript, especially for readers not familiar with this model system.

We have incorporated the suggestion by adding a new figure in the Discussion section (new Figure 5) that shows both the previous model of coevolution and our new understanding of it after this study. We believe this will greatly help the readers who are not familiar with this coevolutionary model system. We also considered moving supplemental figure 1 to the main text but decided not to out of our own stylistic preference. We would like to note that *eLife* online publication makes it very easy to see figures, as they and all the supplements are well-linked with pop-open views embedded within the main text.

• Similarly, the abstract appears intentionally vague and perhaps too broad at times. The authors could attempt to rework this to more clearly explain the co-evolutionary mechanisms that were seen in the study.

We rewrote the second half of the abstract to be more specific.

“During the struggle for survival, populations occasionally evolve new functions that give them access to untapped ecological opportunities. Theory suggests that coevolution between species can promote the evolution of such innovations by deforming fitness landscapes in ways that open new adaptive pathways. We directly tested this idea by using high throughput gene editing-phenotyping technology (MAGE-Seq) to measure the fitness landscape of a virus, bacteriophage λ, as it coevolved with its host, the bacterium *Escherichia coli*. Analyses of the empirical fitness landscape revealed mutation-by-mutation-by-host-genotype interactions that prove coevolution modified the contours of λ’s landscape. Computer simulations of λ’s evolution on a static versus shifting fitness landscape showed that the changes in contours increased λ’s chances of evolving the ability to use a new host receptor. By coupling sequencing and pairwise competition experiments, we demonstrated that the first mutation λ evolved enroute to the innovation would only evolve in the presence of the ancestral host, whereas later steps in λ’s evolution required the shift to a resistant host. When time-shift replays of the coevolution experiment were run where host evolution was artificially accelerated, λ did not innovate to use the new receptor. This study provides direct evidence for the role of coevolution in driving evolutionary novelty and provides a quantitative framework for predicting evolution in coevolving ecological communities.”

• I don't understand the rationale for numbering the supplemental figures in the way the authors have done so. Having sub-supplemental figures for each main figure is not logical and I recommend simply labeling supplemental figures in order of appearance in the main text.

This is an *eLife* formatting specification. The first version of the manuscript we submitted had supplemental figures in the more traditional format, but we were asked by the editorial office to re-submit the manuscript in the current form.

• I really enjoyed the experimental design and approach to disentangling these co-evolutionary dynamics. This is more a comment rather than something to be addressed, but the phylogenetic reconstruction of λ genotypes in Figure 3A shows the emergence of A**C and ABC genotypes from a similar genetic background. How likely do the authors think that the C genotype emergence twice simultaneously and have they instead considered recombination playing a role?

We have speculated about the source of C too and we do not have an answer. Some thoughts: We have observed a striking level of parallel evolution between populations separated in different flasks, so it’s possible that C arose multiple times within the same population. However, λ has an active homologous recombination system and we have observed recombination in these studies between the phage and host-encoded phage genes (Borin et al. PNAS 2021), so it could happen. The MOI of these studies is low – fewer phage than bacteria – so that would suggest there’s little opportunity for coinfection and recombination.